# A Synthetic Polymicrobial Community Biofilm Model Demonstrates Spatial Partitioning, Tolerance to Antimicrobial Treatment, Reduced Metabolism, and Small Colony Variants Typical of Chronic Wound Biofilms

**DOI:** 10.3390/pathogens12010118

**Published:** 2023-01-10

**Authors:** Ammara Khalid, Alan R. Cookson, David E. Whitworth, Michael L. Beeton, Lori I. Robins, Sarah E. Maddocks

**Affiliations:** 1Microbiology and Infection Research Group, Cardiff School of Sport and Health Sciences, Cardiff Metropolitan University, Cardiff CF5 2YB, UK; 2Institute of Biological, Environmental and Rural Sciences, Aberystwyth University, Aberystwyth SY23 3DD, UK; 3Department of Life Sciences, Aberystwyth University, Aberystwyth SY23 3DD, UK; 4Department of Physical Sciences, University of Washington, Bothell, WA 98011-8246, USA

**Keywords:** polymicrobial, antimicrobial, wound infection

## Abstract

Understanding chronic wound infection is key for successful treatment and requires accurate laboratory models. We describe a modified biofilm flow device that effectively mimics the chronic wound environment, including simulated wound fluid, a collagen-based 3D biofilm matrix, and a five-species mixture of clinically relevant bacteria (*Pseudomonas aeruginosa*, *Staphylococcus aureus*, *Escherichia coli*, *Enterococcus faecalis*, and *Citrobacter freundii*). Mixed biofilms were cultured for between 3 and 14 days with consistent numbers of bacteria that exhibited reduced metabolic activity, which increased with a high dose of glucose. *S. aureus* was recovered from biofilms as a small colony variant, but as a normal colony variant if *P. aeruginosa* was excluded from the system. Bacteria within the biofilm did not co-aggregate but formed discrete, species-specific clusters. Biofilms demonstrated differential tolerance to the topical antimicrobials Neosporin and HOCl, consistent with protection due to the biofilm lifestyle. The characteristics exhibited within this model match those of real-world wound biofilms, reflecting the clinical scenario and yielding a powerful in vitro tool that is versatile, inexpensive, and pivotal for understanding chronic wound infection.

## 1. Introduction

Biofilms are estimated to be present in over 78% of chronic wound infections, impairing healing and presenting a clinical challenge to successful treatment [1]. Approximately 2.8 million people in the UK have a chronic wound, leading to annual NHS costs exceeding £5 billion [2]. Chronic wound infection is described as a continuum from initial contamination through to established local or systemic infection. Molecular analyses of chronic wound specimens reveal a diverse polymicrobial community comprising predominantly bacteria of the division Firmicutes and the order Enterobacterales, including aerobes and facultative and strict anaerobes [3].

Chronic wound biofilms are embedded throughout the surface and deeper tissues of the wound bed with non-uniform microbial distribution, which complicates diagnosis and treatment [4]. They exhibit high antimicrobial tolerance which contributes to treatment failure with conventional antibiotic and topical antimicrobial interventions. All antimicrobial wound topicals undergo testing prior to marketing, reliant predominantly on modified disk diffusion and minimum inhibitory concentration (MIC) assays, where they often show success. Despite this, there is little evidence for their use in treating chronic wound infections effectively in practice [5,6].

Numerous in vitro models have been developed for the study of chronic wound biofilms to improve translation into the clinical environment (see [1,7,8] for thorough reviews). Most rely on modifications made to salient static biofilm models to incorporate elements that better mimic the host environment, including simulated wound fluids and three-dimensional tissue-like matrices. Other models have moved closer to the in vivo environment by the inclusion of host–matrix proteins such as collagen and fibrinogen, and a physiologically relevant flow of nutrients [9,10,11,12,13,14]. Whilst each model is useful, most are still limited with regard to clinical relevance, versatility, throughput, ease of accessibility, and associated costs.

Here, we describe significant modifications to our dynamic chronic wound biofilm model encompassing facets that better resemble the chronic wound environment. These have resulted in several features that represent real-world chronic wound biofilms, including reduced metabolism, spatial partitioning, and small colony variants. We also show that stable communities can be maintained for 14 days. Our model confirms that topical antimicrobials can permeate a tissue-like substratum to impair susceptible bacteria with associated recalcitrance and raised tolerance, as seen in real-world chronic wound infections. Collectively, these characteristics make this model a useful tool for realistic antimicrobial testing and confer the potential to study the microbial interactions, population dynamics, and metabolic networks that underpin chronic wound infection.

## 2. Materials and Methods

### 2.1. Bacterial Strains and Culture Conditions

*Staphylococcus aureus* EMRSA-15, *Pseudomonas aeruginosa* ATCC 9027, *Citrobacter freundii* NCTC 6272, *Escherichia coli* ATCC 10418, and *Enterococcus faecalis* ATCC 19433 were maintained on nutrient agar or tryptic soy agar. Selective media for re-isolation from mixed biofilms were UTI Chrome (*E. coli*), Coliform Chrome (*C. freundii*), Slanetz and Bartely (*E. faecalis*), Cetrimide (*P. aeruginosa*), and Baird Parker (*S. aureus*). All media were purchased from Sigma-Aldrich (Gillingham, UK). Each species was cultured in 5 mL Nutrient Broth or Tryptic Soy Broth for 16 h at 37 °C and subsequently equilibrated to 1 × 10^8^ Colony-Forming Units (CFU) mL^−1^ in simulated wound fluid (SWF) base (2.34 mM CaCl_2_·2H_2_O, 3.75 mM KCl, 9.9 mM NaCl) [15]. A 5 mL mixture containing a 1:1 ratio of each species was prepared and used in subsequent biofilm growth experiments.

### 2.2. Preparation of Growth Substrata for Biofilms

Alginate (Protanal, FMC Biopolymer, Ayreshire, UK) was prepared at a concentration of 1.5% *w*/*v* in sterile, deionised water. Aliquots of 1 mL were immersed in 10 mL 1.5% (*w*/*v*) CaCl_2_ for 30 min until solid. Sections of 8 mm were cut from the alginate with a sterile leather punch or biopsy punch and inoculated with 10 μL mixed culture using a sterile needle and incubated at 37 °C for 16–24 h (Appendix A). To embed bacteria in alginate, 5 mL of 3% (*w*/*v*) liquid alginate was mixed with 5 mL of bacterial culture prior to aliquoting into 10 mL 1.5% (*w*/*v*) CaCl_2_, as previously described. To recover bacteria, alginate beads were dissolved in 20 mM EDTA or homogenised in 1 mL phosphate-buffered saline using a glass homogeniser. Agarose–collagen was prepared by adding foetal bovine serum (FBS; heat-inactivated; Pan Bio-Tech, Aidenbech, Germany) to 1 mL of bacterial suspension to achieve a concentration of 3% *v*/*v*, NaHCO_3_ to achieve a concentration of 100 mM, and bovine skin collagen to a concentration of 0.2 mg mL^−1^. An amount of 20 mL of 1.5% (*w*/*v*) agarose was dissolved in an SWF base. This was allowed to cool but not set. Then, 10 mL of agarose mixture was poured into the cell suspension. The agarose–bacteria suspension was briefly mixed by inversion and 780 µL aliquots were dispensed into the wells of a 24-well microtitre plate (Sigma Aldrich, Gillingham, UK) and set at room temperature. Agarose plugs of 8mm were cut from the wells using a sterile leather punch or biopsy punch (Appendix A).

### 2.3. Culture of Bacteria in the Biofilm Flow Device

The biofilm flow device was filled with SWF (SWF base plus 3% *v*/*v* heat-inactivated FBS and 100 mM NaHCO_3_) [15] and connected to a peristaltic pump at a flow rate of 0.322 mL min^−1^. Inoculated matrices were transferred into the wells of the flow system and maintained at 33 °C for the duration of the experiment (Appendix A). At designated time points, biofilms were collected and processed (as described above) for Total Viable Count (TVC) using the Miles and Misra method [16]. Successful perfusion into the agarose–collagen matrices was determined by soaking them in 0.025% (*w*/*v*) methylene blue (Sigma Aldrich, Gillingham, UK) for 10, 20, and 30 min and slicing a cross-section to observe the distribution of the dye.

### 2.4. Metabolic Activity

Biofilms were removed from the flow device using sterile forceps at 1 h, 4 h, 6 h, 24 h, 48 h, and 72 h timepoints, homogenised in 1 mL PBS using a glass homogeniser, and a 1 mL volume was transferred into a 24-well microtitre plate (Sigma Aldrich, Gillingham, UK). An amount of 200 µL of Cell-Titer Blue^®^ (Promega, Southampton, UK) was added to the wells of each plate and incubated at room temperature for 1h, according to the manufacturer’s instructions. Metabolic activity was measured using a Tecan Infinite^®^ plate reader (excitation: 550 nm; emission 600 nm). Controls were planktonic cultures assessed at the same time points (1 h, 4 h, 6 h, 24 h, 48 h and 72 h) and equilibrated to 1 × 10^8^ CFU mL^−1^.

### 2.5. Testing Topical Antimicrobials

Agarose–collagen matrices were prepared, and the biofilm flow device was set up as previously described. An amount of 0.15 g of Neosporin cream (equivalent to 810 μg bacitracin, 520 μg neomycin, and 90 μg polymyxin B per biofilm; Johnson & Johnson, New Brunswick, NJ, USA) or HOCl gel (0.018% HOCl; Briotech Inc., Everett, WA, USA) was added to the surface of each matrix to simulate the application of a topical treatment to a wound. Biofilms were collected at 24 h, 48 h, and 72 h and assessed by TVC. Bacteria were re-isolated in pure culture from mixed biofilms collected at 24 h, 48 h, and 72 h. MIC analysis was undertaken using these pure cultures and neomycin, bacitracin or polymyxin B at stock concentrations of 2 mg mL^−1^. Assays were incubated at 37 °C for 16 h. These were compared to laboratory strains that had not been exposed to Neosporin. There are no EUCAST guidelines for the topical use of these antibiotics, but the following are accepted standards: bacitracin (S ≤ 2 mg L^−1^; R ≥ 2 mg L^−1^), neomycin (S ≤ 8 mg L^−1^; R ≥ 16 mg L^−1^), and polymyxin B (S ≤ 2 mg L^−1^; R ≥ 2 mg L^−1^) [17,18,19]. Agar diffusion assays used individual species of each bacterium equilibrated to an optical density of 0.05 at 650 nm. Lawn plates were prepared by spreading 100 μL of bacteria onto nutrient agar and drying at room temperature for 10–15 min. Holes of 6mm were bored into the agar and filled with 0.15 g of either Neosporin cream or HOCl gel; plates were incubated for 16 h at 37 °C. Zones of inhibition were measured in mm using callipers.

### 2.6. Live-Dead Staining Analysis

Biofilms were tested for viability over 72 h. They were removed from the device, homogenised as described above, and transferred into a 24-well microtitre plate. An amount of 20 µL of LIVE/DEAD^TM^ stain (ThermoFisher Scientific, Abingdon, UK) was prepared according to the manufacturer’s instructions and added to the wells of each plate, which were incubated at room temperature in the dark for 20 min. Viability was determined according to the manufacturer’s instructions using a Tecan Infinite^®^ plate reader (excitation: 485 nm; SYTO9 emission 530 nm; propidium iodide emission 630 nm). For fluorescence microscopy, biofilm plugs were sliced into cross-sections of 1–2 mm depth using a sterile scalpel and stained with LIVE/DEAD^TM^ stain at room temperature in the dark for 20 min, then washed with deionised water. These were secured between a glass slide and coverslip, which was held in place with adhesive. Images were acquired using a Nikon Eclipse 80i fluorescent microscope (Nikon, Surrey, UK) with oil immersion and ×100 lens. SYTO 9 detection (green channel; detects live cells) used a 488 nm excitation and 520 nm emission filter. Propidium iodide detection (red channel; detects dead cells) used a 543 nm excitation and 572 nm emission filter. Volocity^®^ 3D Image Analysis Software (v6.3) (RRID:SCR_002668) was used for image analysis (PerkinElmer Inc., Cambridge, UK).

### 2.7. Transmission Electron Microscopy

All timed steps involving fixatives, washes, ethanol mixtures, and resin infiltration were conducted on a suitable rotator at room temperature in a fume hood unless otherwise stated. Centrifugation was always 5 min at 10,000 rpm using a Hettich Mikroliter D-7200 micro-centrifuge. Whole agarose–collagen biofilms were transferred into 1 mL of a primary fixative, which consisted of 2.5% glutaraldehyde in 0.1 M sodium cacodylate at pH 7.2 (both Agar Scientific Ltd., Stansted, UK). After 30 min fixation, the samples were centrifuged, and the supernatant was discarded. The pellets were re-suspended in another 1 mL of fresh fixative, as above. After 30 min, the previous step was repeated, but samples were re-suspended instead in 1 mL 0.1 M sodium cacodylate wash buffer pH at 7.2. The samples were centrifuged, and the supernatant was discarded. They were re-suspended in 1 mL of a secondary fixative consisting of 1% (*w*/*v*) osmium tetroxide (Agar Scientific Ltd., Stansted, UK) made up in 0.1 M sodium cacodylate buffer at pH 7.2. After 30 min fixation, the samples were centrifuged, and the supernatant was carefully discarded. It was replaced with a quick rinse in 1 mL of wash buffer, as above. After a 5 min rinse, the samples were centrifuged, and the supernatant was discarded. The pellets were re-suspended in another 1 mL of wash buffer. The samples were centrifuged, and the supernatant was discarded. The samples were re-suspended in 100 µL agarose solution (2% ultra-low gelling temperature agarose solution (Sigma Aldrich, Gillingham, UK) was made up in ultra-pure H_2_O, which was dissolved at 50 °C, cooled, and filtered with a 0.22 µm syringe filter; Whatman Ltd., Buckinghamshire, UK) at 25 °C and placed in a refrigerator to gel at 4 °C.

After gelling overnight, the agarose was cut from the Eppendorf tubes with single-sided razor blades and transferred into 1 mL wash buffer in 5 mL glass vials at 4 °C. After 30 min, the gelled agarose pellets were placed in a fresh wash buffer. The samples were then progressed through an ethanol series of 30%, 50%, 70%, 95%, and three changes of 100% for at least an hour. The samples were transferred to a 1:2 mixture of ethanol to LR White Hard Grade (Agar Scientific Ltd., Stansted, UK) resin then a 2:1 mixture of ethanol to resin and finally 100% resin overnight at 4 °C. The resin was then removed and replaced with fresh resin, placed in size 4 gelatine moulds (Agar Scientific Ltd., Stansted, UK), filled up with fresh resin, and polymerised overnight in an oven at 60 °C. Sections 2 µm thick, containing the bacteria, were cut and dried down on drops of 10% ethanol on glass microscope slides. They were stained with AMB stain (Azure II and methylene blue, both Sigma Aldrich, Gillingham, UK) and photographed using a Leica DM6000B microscope. Ultrathin 60–80 nm sections were then cut on a Reichert-Jung Ultracut E Ultramicrotome with a Diatome Ultra 45 diamond knife and collected on Gilder GS2X0.5 3.05 mm-diameter nickel slot grids (Gilder Grids, Grantham, UK) float-coated with Butvar B98 polymer (Agar Scientific Ltd., Stansted, UK) films. All sections were double-stained with uranyl acetate (Agar Scientific Ltd., Stansted, UK) and Reynold’s lead citrate (TAAB Laboratories Equipment Ltd., Aldermaston, UK) and observed using a JEOL JEM1010 transmission electron microscope (JEOL Ltd., Tokyo, Japan) at 80 kV. The resulting images were photographed using a Carestream 4489 electron microscope film (Agar Scientific Ltd., Stansted, UK) developed in Kodak D-19 developer for 4 min at 20 °C, fixed, washed, and dried according to the manufacturer’s instructions. The resulting negatives were scanned with an Epson Perfection V800 film scanner and converted to positive images.

### 2.8. Co-Aggregation

Cultures were prepared as described above and equilibrated to OD 0.7 (A650). Equilibrated cultures were pelleted at 5000 g for 10 min and resuspended in 1 mL coaggregation buffer (1 mM Tris (pH 8.0), 150 mM NaCl, 0.1 mM CaCl_2_·2H_2_O, 0.1 mM MgCl_2_·6H_2_O, 0.02% (*w*/*v*) Na_3_N). Paired co-aggregation used equal volumes of two species mixed in a semi-microcuvette. Turbidity was measured at 0, 30, 90, and 180 min using a spectrophotometer at A650.

### 2.9. Statistical Analysis 

Statistical analysis used ANOVA with a post hoc Tukey’s test, undertaken using Minitab^®^ (v21.1.0; Minitab, LLC, https://minitab.com, 2021). Statistical significance was determined as *p* < 0.05. The standard error of the mean was calculated based on between 3 and 9 biological and technical replicates per experiment.

## 3. Results

### 3.1. Optimising a Five-Species Biofilm for Continuous Culture under Conditions of Flow in a Three-Dimensional Tissue-like Matrix

Initial experiments utilised a static growth system with an alginate matrix [20]. Bacteria were either injected into the solidified matrix or embedded by mixing with liquid alginate prior to gelling. The intent was to optimise a suitable 3D substratum for consistent biofilm growth using a simple system prior to incorporating it into the more complex flow model. Bacteria were recovered by dissolving the alginate in 20 mM EDTA. However, the numbers of *P. aeruginosa* were unexpectedly and significantly low (*p* < 0.05) compared to the other four species (Figure 1A,B), likely due to the inhibitory effect of EDTA [21,22]. Higher numbers of *P. aeruginosa* were recovered when the alginate matrix was mechanically homogenised instead (Figure 1C,D). However, when incorporated into the flow model, the alginate matrix unpredictably dissolved; it is possible that the combination of flow (0.032 mL/min) and salts in the simulated wound fluid (SWF) disrupted the alginate cross-linking, causing it to disintegrate.

Due to the lack of strength of the alginate matrix, we pivoted to agarose. Agarose has been used as a biofilm matrix to provide a 3D substratum for the growth of bacterial biofilms and does not readily dissolve without the addition of an agarase enzyme [23]. Here, we supplemented the agarose with a physiologically relevant amount of collagen (0.2 mg mL^−1^). Agarose–collagen-embedded bacteria cultured under flow (0.032 mL min^−1^) were consistently recoverable at each time point (Figure 2A). *E. faecalis*, known to be a fastidious microorganism, was routinely recovered in significantly (*p* < 0.05) lower numbers than the other four bacteria. At each time point, *S. aureus* was recovered only as small colony variants (SCVs), which is indicative of stress (Figure 2B). Agarose–collagen matrices were perfused with methylene blue for 30 min to assess permeability. Analysis of the matrices after being cut through the middle showed even staining. This is indicative of effective nutrient permeation for bacteria embedded throughout the matrix under flow (Figure 2C).

### 3.2. Low Metabolic Activity Is Observed in Biofilms over 72 h but Stimulated by Diabetic Concentrations of Glucose

Biofilms were collected at 1 h, 4 h, 6 h, 24 h, 48 h, and 72 h and assessed for metabolic activity by the reduction of resazurin to resorufin, compared to an equilibrated planktonic culture. Biofilms appeared less metabolically active than planktonic culture (*p* < 0.05) (Figure 3), despite the consistency in viable bacteria recovered by TVC. High levels of glucose are known to stimulate the growth of dormant cells. Therefore, glucose was added to the SWF at 10 mg mL^−1^ (100 mg/dL; pre-diabetic serum glucose concentration based on random plasma glucose test; World Health Organisation: https://www.who.int/data/gho/indicator-metadata-registry/imr-details/; accessed on 15 May 2022) and 20 mg mL^−1^ (≥200 mg/dL; diabetic serum glucose concentration based on random plasma glucose test; Diabetes UK: https://www.diabetes.co.uk/diabetes_care/blood-sugar-level-ranges.html; accessed on 15 May 2022). At 10 mg mL^−1^, no significant difference (*p* > 0.05) was observed for biofilm metabolism. However, at 20 mg mL^−1^, the metabolic rate of the biofilm was significantly (*p* < 0.05) increased (Figure 3).

### 3.3. Topical Antibiotics and Antiseptics Reduce Bacterial Numbers with Observable Recovery and Tolerance Developing over Time

The commercially available topical antibiotic cream Neosporin was applied at 0 h to the surface of the agarose–collagen matrices containing embedded bacteria. Bacterial enumeration at 24 h, 48 h, and 72 h showed that the numbers of each bacterium, except for *E. faecalis*, were significantly reduced (*p* < 0.05) at 24 h compared to the untreated control (Figure 4A). Significant reductions (*p* < 0.05) in bacterial numbers were maintained at 48 h for *S. aureus* and *E. coli*. Generally, bacterial numbers recovered after 72 h were equivalent to the untreated controls. Agar diffusion assay verified that each bacterium had susceptibility to Neosporin at the dose applied (Table 1).

Subsequent inhibition of growth analysis for each species re-isolated from the biofilm after 72 h was undertaken using neomycin, polymyxin B, and bacitracin, as appropriate (Table 2). The results demonstrated that some but not all the re-isolated, Neosporin-treated bacteria had significantly increased tolerance compared to pure cultures maintained in lab stocks that had never been co-cultured or exposed to neomycin. The exception was *P. aeruginosa*, which appeared more susceptible. For *C. freundii*, both biofilm growth and Neosporin treatment drove increased tolerance for neomycin.

The application of a topical HOCl antiseptic gel previously shown to be effective against *P. aeruginosa* and *S. aureus* had significant (*p* < 0.05) activity against *S. aureus*, *P. aeruginosa*, and *C. freundii* compared to an NaCl vehicle control (Figure 4B). In agar diffusion assays for single species, HOCl inhibited the growth of all five bacteria (Table 1). This suggests that HOCl permeated the collagen–agarose matrix to inhibit some bacteria but that the polymicrobial nature of the biofilm likely conferred protection to otherwise susceptible organisms. The vehicle control and HOCl resulted in an increase in the numbers of *E. faecalis* above those typically observed. The former might be due to halotolerance on the part of *E. faecalis*, and the latter to reduced numbers of *P. aeruginosa* (which is known to inhibit Gram-positive bacteria) consequent to HOCl treatment.

### 3.4. Pseudomonas aeruginosa Is Responsible for the Small Colony Variant Phenotype in Staphylococcus aureus

Using the agarose–collagen flow system, four-species biofilms were cultured for 72 h. *P. aeruginosa* is known to secrete compounds that are inhibitory to *S. aureus* and was therefore excluded. Notably (*p* < 0.05) higher numbers of *E. faecalis* were recovered in the absence of *P. aeruginosa* (Figure 5A). While numbers of *S. aureus* were not significantly higher (*p* > 0.05) without *P. aeruginosa*, they were consistently recovered as a normal variant rather than SCVs. When *P. aeruginosa* was introduced to the four-species biofilm by inoculation at 24 h, *S. aureus* and *E. faecalis* were subsequently significantly reduced in number, and *S. aureus* was only recoverable as an SCV (Figure 5B,C) implying that *P. aeruginosa* impaired growth and induced an SCV phenotype.

### 3.5. Biofilms Can Be Maintained for 14 Days without the Population Crashing

Chronic wound infections can endure for months; however, microbial populations under laboratory culture conditions typically crash after a short period of time, making it difficult to model infection over a relevant timeframe and produce clinical observations. Within the constraints of our experimental set-up, it was possible to culture a sustainable five-species biofilm for 14 days (Figure 6). Bacterial numbers were maintained at a steady state for the duration of this time.

### 3.6. Bacteria Grow as Discrete Species-Specific Aggregates in the 3D Matrix

Biofilms collected at 24 h, 48 h, and 72 h were sectioned into 1–2 mm slices and stained using LIVE/DEAD^TM^ to differentiate between viable and non-viable bacteria, then visualised by fluorescence microscopy. Images showed that at each time point, predominantly viable bacteria were arranged in discrete aggregates (Figure 7A–C). At higher magnification, it was possible to see that aggregates comprised either cocci or rods with no obvious mixing of cell types (Figure 7C). 

TEM analysis of biofilms at 24 h and 72 h confirmed the presence of discrete aggregates of bacteria, as previously observed in fluorescence analysis (Figure 7A–C). Images generated using stacked cross-sections of 72 h biofilms dual-labelled for *E. coli* (15 nm gold labelled antibodies) and *P. aeruginosa* (10 nm gold labelled antibodies) confirm that the observed aggregates contained single species of bacteria (Figure 8). At 72 h, *E. coli*-specific gold-labelled antibodies were detected in channels or “tracks” within the agarose–collagen matrix, suggesting that intra-colony channels had formed through which *E. coli* cells had passed (Figure 9A,B). Paired coaggregation experiments supported these data (Table 3). None of the bacteria coaggregated in SWF, where a percentage decrease in OD greater than 30% indicates coaggregation [24].

## 4. Discussion

Understanding chronic wound infection is limited by a paucity of realistic laboratory models for the sustained growth of multi-species biofilms in a physiologically relevant environment. A realistic model with clinical relevance has value for wound care and diagnostics, antimicrobial stewardship, and advancing fundamental knowledge of microbial interactions in complex biofilm infections. Here, we describe a polymicrobial chronic wound infection model that is sustainable as a biofilm for 14 days with consistent numbers of viable bacteria. The metabolic activity of whole biofilms was significantly diminished, which is characteristic of the biofilm lifestyle in which metabolic remodelling occurs and is seen clinically [25]. Dormant bacteria such as those in biofilms are regarded as persisters and glucose is known to be a growth-promoting signal that kick-starts metabolism in these populations [26]. Our modified system yields this same result, which could also be relevant to understanding the pathophysiology of chronic wounds that occur in diabetics. 

Models that allow the extended growth of chronic wound-like biofilms in vitro often lack key components that effectively mimic the wound environment [10,12,13,27]. The combination of elements in our chronic wound biofilm model includes physiologically relevant flow rate, substrata, media composition, and basal perfusion, making it more representative than other systems currently used to study chronic wound infection. It is likely that slow metabolism combined with a system that removes waste products enables the maintenance of the biofilm population, and we anticipate it could be sustained beyond 14 days. We are not aware of other biofilm models that permit the long-term growth of biofilms comprising more than two to three species of bacteria. 

Given the longevity of chronic wound infection, our model has the potential to enable microbial community development in a chronic wound-like environment to be studied over a more representative timeframe, considering that most wounds are regarded as chronic if they fail to heal within a three-month timeframe [28]. This affords the possibility to capture microbial adaptations to the environment, community members, and antimicrobials, with valuable ramifications for chronic wound management. We routinely recovered *S. aureus* as SCV morphotypes that were absent when *P. aeruginosa* was excluded from the model. This concurs with the accepted paradigm whereby *P. aeruginosa* induces an SCV phenotype in *S. aureus* and aligns with laboratory observations of chronic wound specimens [29]. At 72 h, tracks formed by the apparent movement of *E. coli* were observed within the gel matrix. Despite our system comprising three different motile bacteria, this supports prior studies describing intra-colony channels in single-species biofilms of *E. coli* [30].

Chronic wound biopsies reveal that bacteria do not grow homogeneously in mixed biofilm infections but form discrete, species-specific aggregates in distinct regions of the tissue [4]. For example, *S. aureus* aggregates near the surface, with *P. aeruginosa* in deeper tissues. To replicate the real-world environment, agarose has been used to culture biofilms relevant to both infection and the food industry [11,31,32,33]. *S. aureus* embedded in agarose is known to develop aggregates in the same way as biofilms in chronically infected lung tissue, indicating clinically relevant bacterial colonisation [23]. Similarly, in this study, despite mixed cultures being homogenous when the agarose–collagen matrix was prepared, discrete aggregates of single species were observed, akin to chronic wounds. Fluorescent microscopy revealed that aggregates were composed of predominantly viable microorganisms, established as single species by TEM. The spatial partitioning of bacteria within our model might underpin the sustained, stable population we observed, in contrast to competitive interactions that have been previously reported using strategies that do not allow for biofilm development or three-dimensional partitioning [34,35,36]. Interestingly, the negligible co-aggregation we observed inferred that this was not a requirement for biofilm development, in contrast to oral bacteria, where aggregation underpins the formation of plaque [37]. The differing biogeography of the oral cavity compared to wound tissues could explain this disparity and highlights the importance of accurately replicating the host environment in vitro to properly understand biological processes. 

Crabbé et al. [38] recently noted that the diminished metabolic activity of biofilm bacteria contributes to their elevated tolerance to antibiotics. Using Neosporin, a mixture of broad-spectrum antibiotics, we saw an initial reduction in bacterial load with subsequent recovery, mirroring the usual clinical course of events where wound infection becomes recalcitrant to treatment. In some cases, Neosporin exposure had a greater impact on antibiotic tolerance than biofilm lifestyle alone. It is possible that re-isolated bacteria “lost” any tolerance the biofilm lifestyle might have conferred, due to the necessity for liquid culture during MIC analysis. The unexpected increase in the susceptibility of *P. aeruginosa* to polymyxin B and neomycin seems contrary to current understanding but has been similarly observed in two-species biofilms of *P. aeruginosa* and *S. aureus* treated with aminoglycoside antibiotics [39]. Our model also proved suitable for testing novel antimicrobial wound topicals. HOCl, a broad-spectrum antimicrobial, significantly reduced the numbers of susceptible biofilm members, including those most often persistent in chronic wounds. This contrasted with agar diffusion assays in which each species exhibited susceptibility to HOCl, reiterating the need for applicable biofilm models to translate observations from bench to clinic.

## 5. Conclusions

We have modified a polymicrobial biofilm flow system to simulate a chronic infected wound and have done so in such a way that the bacteria therein exhibit characteristics that match those of real-world chronic wound biofilms. This yields a powerful in vitro tool that is both versatile and inexpensive. Accurate, reliable, and clinically translatable infection models are pivotal for understanding chronic wound infection and for the development of treatments. This model, therefore, has the potential for the high throughput testing of topical antimicrobials in a realistic environment, with a prospective impact on the outcomes of chronic wound treatment. Despite the modifications to our model, it remains to be an in vitro platform, lacking key biotic host elements that are present in vivo. Technologies exist for the co-culture of mammalian cells and bacteria, and integrating these into our model in future will enable us to unravel pertinent host–pathogen interactions that contribute to chronic wound infection. 

## Figures and Tables

**Figure 1 pathogens-12-00118-f001:**
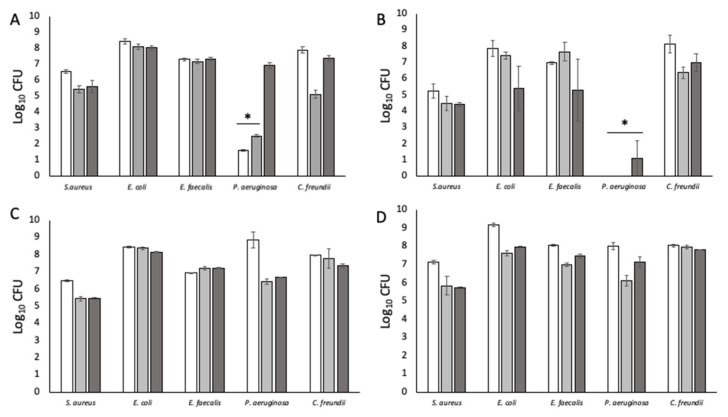
Numbers of bacteria recovered from alginate matrices under different experimental conditions. (**A**) Bacteria inoculated into alginate and recovered by dissolving with 20 mM EDTA. (**B**) Bacteria embedded in alginate and recovered by dissolving with 20 mM EDTA. (**C**) Bacteria injected into alginate and recovered by homogenising. (**D**) Bacteria embedded in alginate and recovered by homogenising. White bars: 24 h; grey bars: 48 h; dark grey bars 72 h. (*) Indicates statistically significant differences (*p* < 0.05) in bacterial count. n = 9.

**Figure 2 pathogens-12-00118-f002:**
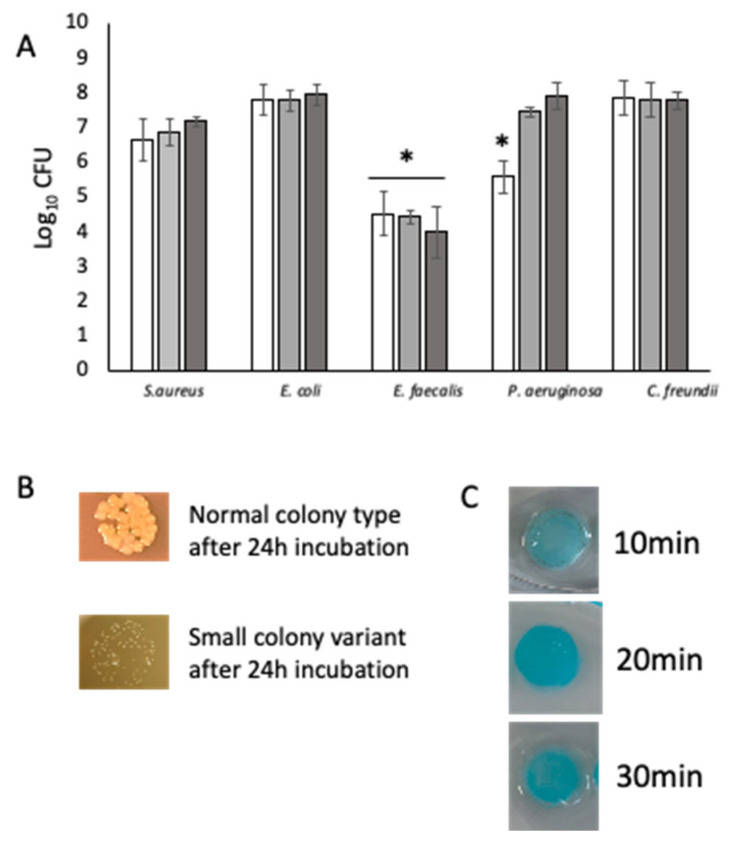
Five-species biofilms cultured for 72 h under flow conditions. (**A**) Each of the five species recovered from homogenised biofilms at 24 h, 48 h, and 72 h. n = 9. (**B**) *S. aureus was* recovered as small colony variants following co-culture in the biofilm model. Top: example of *S. aureus* normal colony type; bottom: example of *S. aureus* SCV recovered from the biofilm model. (**C**) Diffusion of methylene blue through the agarose–collagen matrix assessed at 10 min, 20 min, and 30 min demonstrate nutrient permeation for embedded bacteria under flow. (*) Indicates statistically significant differences (*p* < 0.05) in bacterial count. White bars: 24 h; grey bars: 48 h; dark grey bars 72 h. (*) Indicates statistically significant differences (*p* < 0.05) in bacterial count.

**Figure 3 pathogens-12-00118-f003:**
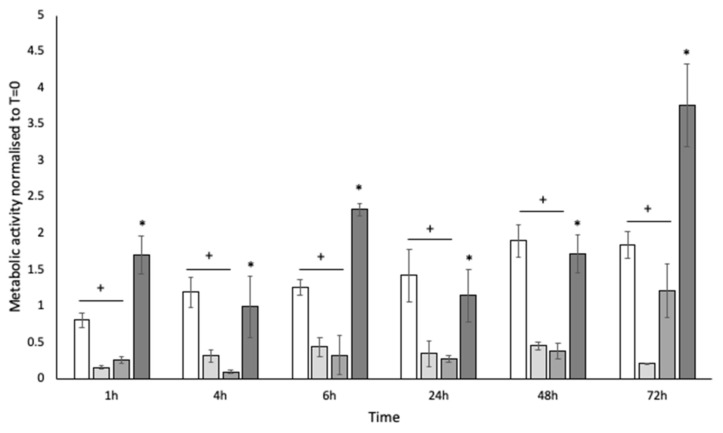
Metabolic activity of whole homogenised biofilms assessed by the reduction of resazurin at 1 h, 4 h, 6 h, 24 h, 48 h, and 72 h. White bars: planktonic culture; pale grey bars: standard SWF (no glucose); mid-grey bars 10 mg mL^−1^ glucose; dark grey bars: 20 mg mL^−1^ glucose. (+) Indicates a statistically significant reduction in metabolic activity compared to planktonic culture; (*) indicates a statistically significant increase in metabolic activity compared to the no-glucose control and 20 mg mL^−1^ glucose. n = 6.

**Figure 4 pathogens-12-00118-f004:**
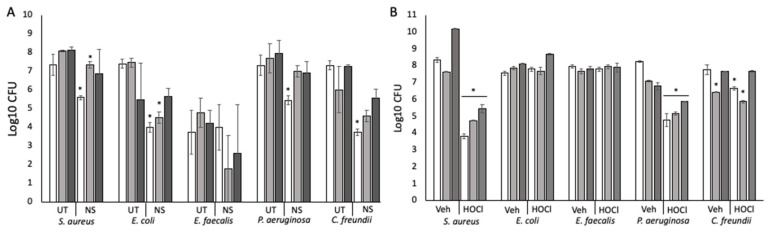
Treatment of biofilms with a topical antibiotic and a topical antiseptic. (**A**) Biofilms treated with Neosporin for 72 h. UT: untreated biofilm; NS: Neosporin-treated biofilm. (**B**) Biofilms treated with HOCl gel or a vehicle control (NaCl) gel. Veh: biofilms treated with the vehicle control; HOCl: HOCl-treated biofilms. White bars: 24 h post-treatment; mid-grey bars: 48 h post-treatment; dark grey bars: 72 h post-treatment. (*) Indicates statistically significant differences (*p* < 0.05) in bacterial count. n = 9.

**Figure 5 pathogens-12-00118-f005:**
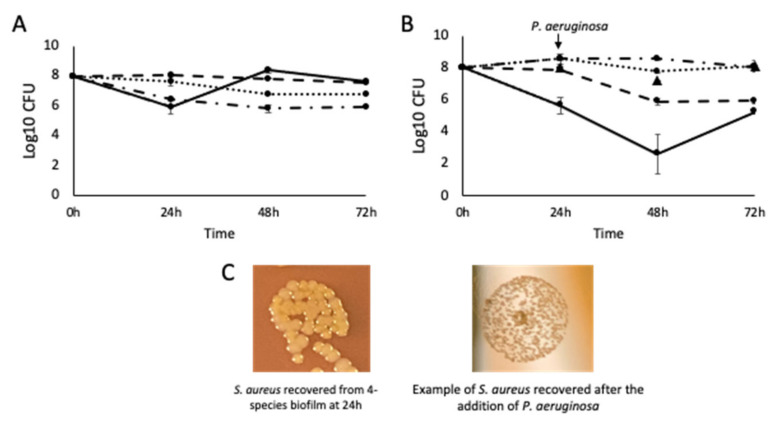
The effect of *P. aeruginosa* on the growth characteristics of *S. aureus* and *E. faecalis*. (**A**) Four-species biofilm excluding *P. aeruginosa.* n = 9. (**B**) Four-species biofilms with the addition of *P. aeruginosa* at 24 h. n = 9 (PA: Log10 CFU at 48 h and 72 h, indicated by black triangles) (**C**) The change in morphotype of S. aureus after the introduction of *P. aeruginosa* to the biofilm at 24 h. *S. aureus*: solid line; *E. coli*: dotted line; *E. faecalis*: short dashes; *P. aeruginosa*: long dashes; *C. freundii*: dots and dashes.

**Figure 6 pathogens-12-00118-f006:**
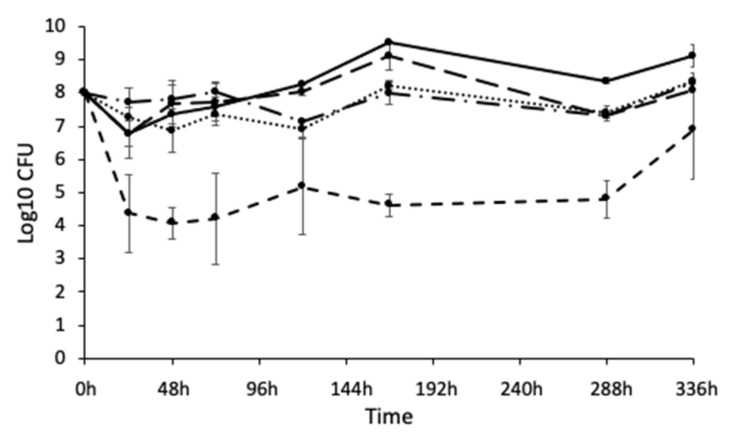
Five-species biofilm cultured at 33 °C under flow for 336 h (14 days). The population remained relatively stable over time, with *E. faecalis* consistently the least numerous. n = 9. *S. aureus*: solid line; *E. coli*: dotted line; *E. faecalis*: short dashes; *P. aeruginosa*: long dashes; *C. freundii*: dots and dashes.

**Figure 7 pathogens-12-00118-f007:**
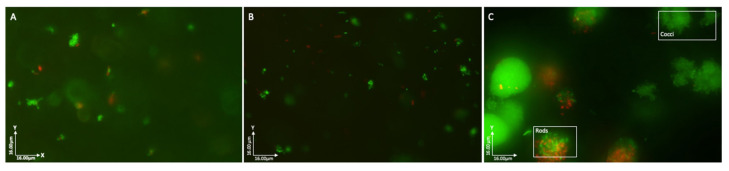
Fluorescent microscopy of biofilms stained with LIVE/DEAD^TM^ (SYTO9 and PI). (**A**) 24 h biofilm; (**B**) 48 h biofilm; (**C**) 72 h biofilm. Images show predominantly viable bacteria arranged in discrete aggregates comprised of bacteria with similar cell morphologies (rod or coccus).

**Figure 8 pathogens-12-00118-f008:**
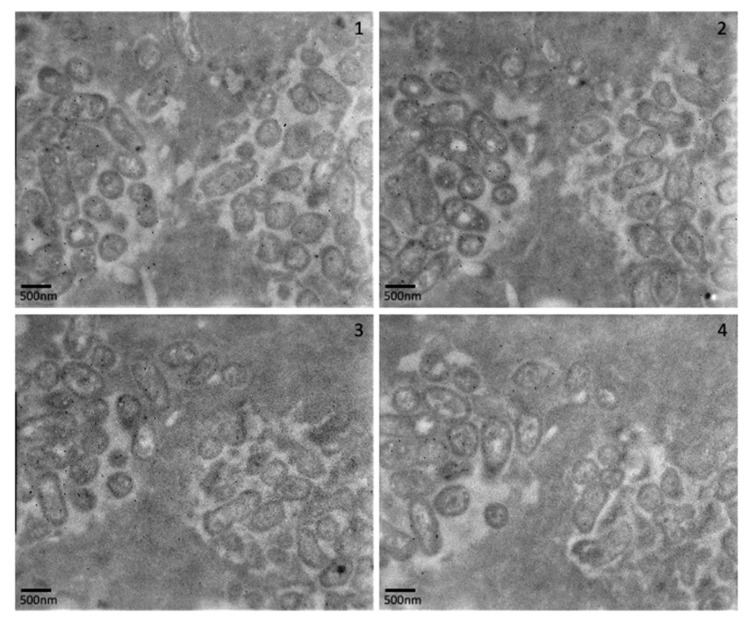
Transmission electron microscopy of biofilm sections (top–bottom: 1–4) with gold-labelled antibodies specific to *E. coli* (15 nm) and *P. aeruginosa* (10 nm). In each image, *E. coli* is seen in aggregates on the left of the images, and *P. aeruginosa* is seen in aggregates on the right of the images.

**Figure 9 pathogens-12-00118-f009:**
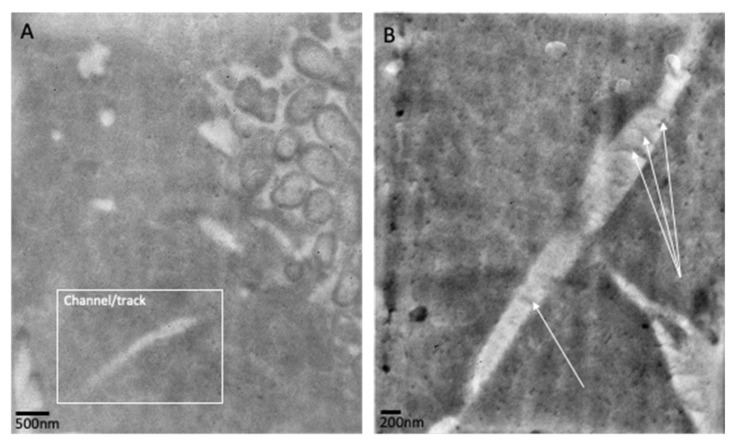
Transmission electron microscopy of mixed biofilms (at 72 h) labelled with *E. coli*-specific gold-labelled antibodies. (**A**) An aggregate of *E. coli* is visible on the right of the image with a channel/track close by. (**B**) Channel/track through the collagen–agarose matrix stained with *E. coli*-specific gold-labelled antibodies. White arrows highlight striations that may signify the direction of movement.

**Table 1 pathogens-12-00118-t001:** Agar diffusion assay for Neosporin and HOCl gel tested against lawns of individual species of bacteria.

	Zone of Inhibition (mm)	
	Neosporin	HOCl
*S. aureus*	21.6 [+/− 1.15]	14.3 [+/− 1.52]
*E. coli*	25.3 [+/− 0.57]	22.3 [+/− 0.57]
*E. faecalis*	16 [+/− 1]	12.6 [+/− 1.15]
*P. aeruginosa*	22.3 [+/− 3.21]	13 [+/− 1]
*C. freundii*	25.6 [+/− 1.52]	15.3 [+/− 1.15]

**Table 2 pathogens-12-00118-t002:** Growth inhibition analysis for neomycin, polymyxin B, and bacitracin to assess the impact of co-culture and co-culture + antibiotic on subsequent antibiotic susceptibility.

		Bacitracin	Neomycin	Polymyxin B
		μg Dose per Well Required to Inhibit Growth		
*S. aureus*	Lab stock (pure culture)	200	6.2 [+/− 0.5]	nt
	Untreated, re-isolated from biofilm	200	6.2 [+/− 0.8]	nt
	Neosporin-treated, re-isolated from biofilm	200	12.5 [+/− 0.4] *	nt
*E. coli*	Lab stock (pure culture)	nt	<0.15	<0.15
	Untreated, re-isolated from biofilm	nt	<0.15	<0.15
	Neosporin-treated, re-isolated from biofilm	nt	3.1 [+/− 3.0]	50 [+/− 2.0] *
*E. faecalis*	Lab stock (pure culture)	100	100	nt
	Untreated, re-isolated from biofilm	75 [+/− 5.0]	100	nt
	Neosporin-treated, re-isolated from biofilm	200	100	nt
*P. aeruginosa*	Lab stock (pure culture)	nt	6.2 [+/− 1.7] *	50 [+/− 2.2] *
	Untreated, re-isolated from biofilm	nt	<0.15	<0.15
	Neosporin-treated, re-isolated from biofilm	nt	<0.15	<0.15
*C. freundii*	Lab stock (pure culture)	nt	<0.15	<0.15
	Untreated, re-isolated from biofilm	nt	3.1 [+/− 0.1] *	<0.15
	Neosporin-treated, re-isolated from biofilm	nt	3.1 [+/− 0.5] *	<0.15

nt = not tested (would not be used/relevant for this bacterium). (*) Indicates a statistically significant (*p* < 0.05) difference.

**Table 3 pathogens-12-00118-t003:** Paired co-aggregation of bacteria in simulated wound fluid. Values exceeding a 30% reduction in OD are regarded as co-aggregation.

	Co-Aggregation Pairs	% Reduction in OD at 30 min	% Reduction in OD at 90 min
*P. aeruginosa*	*S. aureus*	24.30	28.60
	*E. faecalis*	24	32
	*C. freundii*	17.14	20
	*E. coli*	17.14	24.20
*S. aureus*	*E. faecalis*	7.14	10
	*C. freundii*	2.85	7.14
	*E. coli*	2.85	5.7
*C. freundii*	*E. faecalis*	4.20	5.7
	*E. coli*	7.14	7.14
*E. faecalis*	*E. coli*	14.20	18.20

## Data Availability

The data presented in this study are available on request from the corresponding author.

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
