# Peer review of "A Synthetic Polymicrobial Community Biofilm Model Demonstrates Spatial Partitioning, Tolerance to Antimicrobial Treatment, Reduced Metabolism, and Small Colony Variants Typical of Chronic Wound Biofilms"

_pathogens, 2023, doi:10.3390/pathogens12010118_

Round 1
Reviewer 1 Report
Reviewer Report
Comments and Suggestions for Authors - pathogens-2127872
Congratulations to the authors for the excellent work. The manuscript is clear and well-written and definitely it deserves to be published. However, I believe that there is room for a little improvement.
The Material and Methods and Results sections really need to be improved. I recommended that the authors realized a new figure explaining the procedure involving the preparation of the biofilm’s platform for biofilm growth (subsection 2.2) and the in vitro simulation in the biofilm flow device (subsection 2.3). However, it is not mandatory.
I will endorse the revised version of the manuscript after some minor revisions, which I put in my comments below.
Minor comments
Introduction
Line 49- Please put “in vitro” in italics form.
Line 53- Please put “in vivo” in italics form.
Materials and Methods
Line 76- Please add the full name “colony-forming unit” before the abbreviation CFU.
Lines 93-94- Just for curiosity (it is not a correction), why exactly are “780μl aliquots dispensed into the wells of a 24-well microtitre plate”? Is there a particular reason for this exact volume?
Lines 101-102- Again, (it is not a correction), why 33ºC? “Inoculated matrices were transferred into the wells of the flow system and maintained at 33oC for the duration of the experiment.” Is there a particular reason for this temperature? Should not be 37ºC?
Lines 103-104- Please replace “Miles Misra method” with “Miles and Misra method”.
Section 2.4 lines 109-114- Please clarify to the Readers how the authors “Biofilms were removed from the flow device, homogenised, and transferred into a 24 well microtitre plate” and further describe “Controls were equivalent planktonic cultures at each time point, equilibrated to 1x108 CFU ml-1.”, indicating exactly the analysed time points.
Lines 117-118- Please replace “as described” with “as previously described”.
Line 122- Please add full name total viable count before “TVC”. The authors wrote it on page 7 line 267, but here is the first citation.
Line 125- Please replace “Neomycin” with “neomycin”, as previously written in line 123. Also, rectify again at line 132 checking in the remaining manuscript for the same discrepancy.
Line 129- Please confirm “bacterium equilibrated to OD 0.05 at A650”. It should be the optical density of 0.05 measured at 600 nm, right?
Lines 142-143- Please clarify with more details for the Readers how “For fluorescence microscopy, biofilm plugs were sliced into segments of 1-2mm depth.”.
Line 145- Please rectify “Nikon Eclipse 80i fluorescent micro- scope”.
Lines 148-149- Please add the version of Volocity software and cite its reference.
Lines 208- Please add the version of Minitab software and cite its reference.
Results
Line 219- Please replace “(Figure 1a and 1b),” with “(Figure 1A and 1B),”. The same systematic typo error is found in the remaining figures. Please also rectify it in the remaining text of the manuscript.
Page 6- Figure 2 the bar colors white, light grey, and dark grey represent culture growth at 24, 48, and 72 hours (as previously indicated in Figure 1), right? Please add the bar colors’ legend in Figure 2.
Line 284- Please rectify the legend of Figure 3 “… 10mg/ml glucose. n=6”.
Line 286- Please remove the final dot/point of the title in section 3.3.
Lines 304-305- Please put the title of Table 1 before the respective table on line 302, as indicated in the author guidelines or MDPI Word template. Also, Table 1 was inserted in the manuscript as an image instead of the real table, please rectify it. The same systematic errors (title and image instead of the actual table) are found in the remaining tables. Please also rectify it in the remaining text of the manuscript.
Lines 322-323- Please clarify the sentence: “The former might be due to halotolerance, and the latter to reduced numbers of P. aeruginosa, which is known to inhibit Gram-positive bacteria”.
Lines 340-341-Please put “S. aureus” in the italic form.
Please move Figure 5 to subsection 3.4.
Please move Figure 6 to subsection 3.5.
Line 354 and 387- Please rectify the sentence “using LIVE/DEADTM for confocal microscopy” and the title/legend of Figure 7 “Confocal laser scanning microscopy of biofilms”
The Nikon Eclipse 80i fluorescent microscope, as well indicated by the authors in subsection 2.6 at line 142 “For fluorescence microscopy, biofilm”, was used for fluorescence microscopy analysis, and the images that I see in Figure 7 also seem of fluorescence microscopy analysis and it seems not an evaluation by confocal laser scanning microscopy. Please rectify this misunderstanding error for the Readers in Figure 7 and the text of subsection 3.6.
Lines 357-358- Please rectify the sentence “TEM analysis of biofilms at 24h and 72h confirmed the presence of discrete aggregates of bacteria (Figure 7a, 7b, and 7c)” as “TEM analysis of biofilms at 24h and 72h confirmed the presence of discrete aggregates of bacteria, as previously observed in fluorescence analysis (Figure 7a, 7b, and 7c)”.
Page 11 and 12- Please improve the resolution of the TEM images of Figures 8 and 9, more precisely, the scale text.
Discussion
Line 419- Please check the letter type at “[26].”, it seems different from the remaining text.
The discussion seems too short with only one page and 8 references. I would invite the authors to improve this section by comparing their results with previous studies reporting methodological data of these pathogens’ biofilms or/and real data of medical practice observations. I recommend the authors add information about fluorescence microscopy and TEM analyses, which is lacking in the Discussion section.
Conclusions
Line 465- Please put “in vitro” in italics form.
I would invite the authors to recognize the shortcomings of the present study and their goals for future studies.
Again, congratulations to the authors for their excellent work. It was a pleasure to read your study. The manuscript is clear and well-written with exhaustive work done by the authors, and it deserves to be published. However, the manuscript really needed to be improved. I will endorse publication after the authors realized the previous corrections/suggestions.
Reviewer 2 Report
I consider the article of interest, and it is well presented. The authors modified the polymicrobial biofilms above, intending to simulate an infected chronic wound, achieving a proposal to eliminate superimposed biofilms. This is important since it will be possible to measure the effectiveness of specific compounds in removing biofilms.
Reviewer 3 Report
Manuscript has been written very well covering almost all aspect. There are few minor revisions.
1. Give the full form of SWF (line 91) and TVC (line 122). Full forms have been used somewhere ahead in the manuscript.
2. Lines 236-238. Agarose has been used as a biofilm matrix to provide a 3D substratum for the growth of bacterial biofilms and does not readily dissolve without the addition of agarase. The authors have written ‘3D agarose substratum does not readily dissolve without addition of agarose” Is it agarose or something else. It means agarose is dissolving the agarose. Please clarify
3. Metabolic activity of biofilms assessed by reduction of resazurin at various time intervals. It has been just mentioned in Figure 3. Elaborate metabolic activity in terms of reduction in resazurin in lines 262-275.
